# Correlations as a resource in quantum thermodynamics

Facundo Sapienza[1], Federico Cerisola [1,2] & Augusto J. Roncaglia[1,2]

The presence of correlations in physical systems can be a valuable resource for many quantum information tasks. They are also relevant in thermodynamic transformations, and their creation is usually associated to some energetic cost. In this work, we study the role of correlations in the thermodynamic process of state formation in the single-shot regime, and find that correlations can also be viewed as a resource. First, we show that the energetic cost of creating multiple copies of a given state can be reduced by allowing correlations in the final state. We obtain the minimum cost for every finite number of subsystems, and then we show that this feature is not restricted to the case of copies. More generally, we demonstrate that in the asymptotic limit, by allowing a logarithmic amount of correlations, we can recover standard results where the free energy quantifies this minimum cost.

---

[1] Departamento de Física, FCEyN, Universidad de Buenos Aires, Ciudad Universitaria, 1428 Buenos Aires, Argentina. [2] Instituto de Física de Buenos Aires, CONICET, Ciudad Universitaria, 1428 Buenos Aires, Argentina. Correspondence and requests for materials should be addressed to F.C. (email: cerisola@df.uba.ar) or to A.J.R. (email: augusto@df.uba.ar)

Quantum thermodynamics is a growing field aiming to extend thermodynamics to the limit of few number of systems in the quantum domain[1,2]. This quest has been motivated by the theoretical interest in understanding the fundamental limitations of thermodynamic transformations, and from a practical point of view it has been driven by the current technologies that allow to reach an incredible level of control of individual quantum systems. Among the different approaches that have been put forward to analyze thermodynamics in this regime, a recent perspective to study non-equilibrium transformations of small number of systems in contact with a thermal bath, the so-called resource theory of thermodynamics[3–6], has gained a lot of interest[7–24]. This framework captures the fundamental concepts of thermodynamics with an operational approach to physics[25]: by defining a set of operations an agent is allowed to perform on a physical system, it characterizes the set of attainable transformations. The resource-theoretic approach to thermodynamics, while consistent with classical thermodynamics, has interesting properties that depart significantly from the standard framework. In fact, in the single-shot regime thermodynamic transformations must satisfy a family of constraints[26], including the standard second law as a particular case. Furthermore, it naturally leads to a fundamental notion of irreversibility, since in general the amount of deterministic work required to perform a given transformation is greater than the work that can be drawn from the reverse process[5,6].

One of the main challenges in this field is to elucidate the role of properties such as quantum coherences and correlations in thermodynamic transformations. Several works address the influence of coherence[12,14,15,27,28] and correlations[7,19,29–38] in thermodynamic transformations in different scenarios. In general, the creation of correlations is associated to some energetic cost and strategies to optimally extract work from them have been put forward[33–38]. On the other hand, it has been demonstrated that in the single-shot regime, by allowing auxiliary correlated catalytic systems[7] or correlations with catalytic systems[19], it is possible to enlarge the set of achievable transformations.

In this article, we study how inner correlations can affect certain fundamental processes that take place in contact with a thermal reservoir. In particular, we consider the processes of state formation and work extraction in the single-shot regime. That is, a Gibbs state is transformed into some out-of-equilibrium state using deterministic work, and deterministic work is drawn from an inverse transformation. We start our analysis by concentrating on the work of formation of a finite set of locally identical quantum systems. We find that by allowing correlations in the final state this energetic cost can be reduced (Fig. 1). This is in strike contrast with standard scenario where arbitrary large fluctuations of work are allowed, and the creation of correlations requires some extra energy. While for uncorrelated copies most of these processes are shown to be irreversible, here we show that the degree of reversibility in the correlated scenario increases with the number of copies. Then, we show that in the asymptotic limit the optimal collective process can be accomplished with correlations per particle that are vanishing small, and the work of formation per particle equals the free energy difference. Finally, we generalize these results for an arbitrary set of local systems.

## Results

**Overview.** In standard thermodynamics, the transformations between states that occur in contact with a thermal reservoir are governed by the Helmholtz free energy

$$F(\rho) = \langle E(\rho) \rangle - k_{\mathrm{B}} T S(\rho), \qquad (1)$$

where $\langle E(\rho) \rangle$ is the mean energy of the system in state $\rho$, $S(\rho)$ is

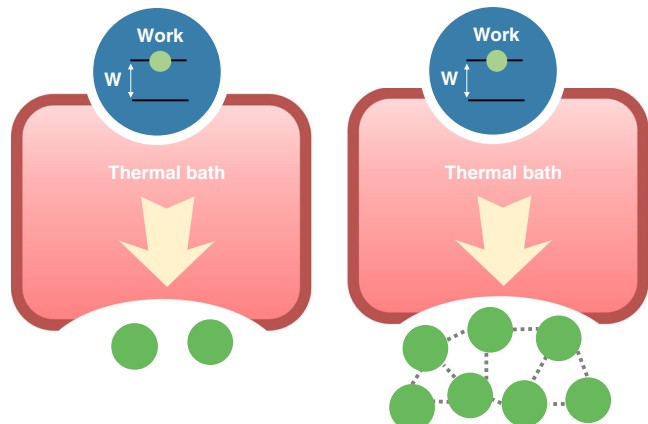

**Fig. 1** Correlations as a resource. In the single-shot regime, the creation of $N$ correlated copies of a given state has a smaller work cost than creating $N$ independent copies. In the asymptotic limit, this energetic cost per copy converges to the standard non-equilibrium free energy difference

the entropy, $k_{\mathrm{B}}$ is Boltzmann's constant, and $T$ is the temperature of the thermal reservoir. Transitions between states are allowed provided that the free energy of the final state is lower than at the beginning. In fact, the difference in free energy is equal to the amount of average work that can be extracted during the process, and is also equal to the work that should be invested in the reverse process. This formulation was developed for macroscopic systems where, due to the large number of particles, energy fluctuations become negligible. On the other hand, if one wishes to understand thermodynamic transformations of a small number of non-equilibrium systems, the size of these fluctuations become important as they could be of the order of the value of work. Recently, an approach that addresses thermodynamic transformations in this regime has been developed, and conditions on state transitions have been identified[3–6]. Below, we briefly introduce the formalism to study thermodynamic transformations in the single-shot regime, and review the main results relevant to this work.

At the core of the theory is the identification of a set of allowed operations, which model the most general transformation in this framework[4,5,26]. Let us consider a system with Hamiltonian $H_{\mathrm{S}}$ and an arbitrary thermal reservoir R in a Gibbs state $\tau_{\mathrm{R}} = e^{-\beta H_{\mathrm{R}}} / \mathrm{tr}[e^{-\beta H_{\mathrm{R}}}]$ with Hamiltonian $H_{\mathrm{R}}$, and $\beta = 1/k_{\mathrm{B}}T$. System and reservoir are allowed to interact via a unitary evolution $U$ that preserves the total energy $[U, H_{\mathrm{S}} + H_{\mathrm{R}}] = 0$, and then it is possible to perform a partial trace over S and R. Given the system in an initial state, the allowed transformations are called thermal operations and they define a set of reachable states. In contrast to other frameworks, where just the conservation of the mean energy is imposed[39], these conditions give a strong conservation of energy (first law of thermodynamics). Thus, given two states $\rho$ and $\sigma$, we say that $\sigma$ can be reached from $\rho$, $\rho \rightarrow \sigma$, if there exists a thermal operation that implements such transformation. A necessary condition for thermodynamic state transitions is called thermo-majorization[5], that is sufficient for diagonal states, i.e., $[\rho, H_{\mathrm{S}}] = 0$, which is also the case we will consider here. Although the thermal operations appear potentially very complex, since they allow any energy conserving interaction between state and bath, it has been shown that they can always be achieved as sequences of elemental operations that have a simple form and physical interpretation[40].

More generally, one can consider transformations that also allow the presence of an additional system that acts as a catalyst

of the transformation, and is returned in the same state. These type of transformations enlarge the set of reachable states, and the necessary and sufficient conditions for diagonal states can be written as an infinite set of inequalities[26]. In this case, a transformation from an initial state $\rho$ to a final state $\sigma$ can be done provided $F_\alpha(\rho) \geq F_\alpha(\sigma)$ for all $\alpha \in \mathbb{R}$, where $F_\alpha$ are the $\alpha$-free energies defined in terms of Rényi divergences $D_\alpha(\rho||\tau_S)$ as $F_\alpha(\rho) = k_B T D_\alpha(\rho||\tau_S) - k_B T \log Z_S$, with $\tau_S = e^{-\beta H_S}/Z_S$ the thermal state of the system. Thus, this is the family of inequalities that govern thermodynamic transformations in this regime[26]. The standard second law is contained as a particular case for $\alpha = 1$.

In the single-shot scenario, the notion of deterministic work can be considered by introducing an auxiliary two-level system $W$ with Hamiltonian $H_W = W|W\rangle\langle W|_W$, called work qubit or wit, that acts as a battery which can store or inject energy into the system[5]. In particular, we will be interested in the energetic cost of obtaining a state $\rho$ out from a thermal state. This work cost can be evaluated by studying the following transformation:

$$\tau_S \otimes |W\rangle\langle W|_W \rightarrow \rho \otimes |0\rangle\langle 0|_W. \qquad (2)$$

The smallest possible value of such $W$ is defined as the work of formation[5], and gives the minimum amount of deterministic work required in the transformation. For diagonal states it is given by

$$W_{form}(\rho) = k_B T D_\infty(\rho \| \tau_S), \qquad (3)$$

which is also equal to $k_B T \log \max_{E,g}\{\lambda_{E,g} e^{\beta E} Z_S\}$, where $\lambda_{E,g}$ are the eigenvalues of $\rho$, $g_S(E)$ the degeneracy, and $g = 1, ..., g_S(E)$. Notice that the work of formation is in general greater than the free energy difference. Similarly, one can define the extractable work as the maximum work that can be stored in the work qubit starting from a state $\rho$, and its expression for diagonal states is given by[5]

$$W_{ext}(\rho) = k_B T D_0(\rho \| \tau_S). \qquad (4)$$

The addition of a catalyst to the process of state creation or work extraction does not modify these values of work. Finally, let us mention an important feature of this theory that is also relevant to our work: in general the extractable work is smaller than the work of formation, thus there is an inherent irreversibility in thermodynamic transformations in this regime[5,26]. However, when correlated catalysts are allowed, the transformations become ruled by just the usual free energy difference[7,19].

**Work of formation of correlated copies.** Let us consider a situation where a finite set of particles is prepared in the same reduced diagonal state $\rho$. This could be for instance the first step of a given task. What is the minimum work cost of producing such $N$-partite ensemble if one is able to interact with a thermal reservoir? There are many multipartite states compatible with this situation, since it is only defined by some reduced state and number of particles, but these states have a different work cost. If we allow arbitrary large fluctuations of work[11], creating a correlated state $\rho^{(N)}$ out of a thermal one is useless. The average work cost associated to correlated copies of $N$ systems with Hamiltonian $H_S$ is given by the standard non-equilibrium free energy difference $\langle W \rangle \equiv F(\rho^{(N)}) - F(\tau_S^{\otimes N})$ which can be expressed as

$$\langle W \rangle = N\Delta F(\rho) + k_B T \mathcal{I}(\rho^{(N)}), \qquad (5)$$

where $\Delta F(\rho) = F(\rho) - F(\tau_S)$, and

$$\mathcal{I}(\rho^{(N)}) \equiv D_1\left(\rho^{(N)} \| \rho^{\otimes N}\right) \qquad (6)$$

is a measure of the total correlations[41], with $D_1(\cdot\|\cdot)$ the relative entropy. This average work cost has two components: the energy

required to obtain $N$ uncorrelated copies $N\Delta F(\rho)$ and the energy associated to the correlations which is also positive. Therefore the above expression tells us that correlations are costly, if unbounded fluctuations of work are allowed, the work cost of this task cannot be reduced by creating correlations between subsystems. In what follows, we will show that in the single-shot scenario a collective action provides an advantage, and in fact the minimum work cost of this task is achieved with correlated copies.

Let us consider $N$ identical $D$-dimensional quantum systems S with Hamiltonian $H_S$. Given a reduced state $\rho = \sum_{d=1}^D p_d |E_d\rangle\langle E_d|$, we are interested in studying the following transformation:

$$\tau_S^{\otimes N} \otimes |W\rangle\langle W| \rightarrow \rho^{(N)} \otimes |0\rangle\langle 0|, \qquad (7)$$

where with an amount of deterministic work $W$ a multipartite state $\rho^{(N)}$ is created, subject to the local condition

$$\text{tr}_{-i}(\rho^{(N)}) = \rho \qquad \forall i = 1, 2, \dots N, \qquad (8)$$

with $\text{tr}_{-i}(\cdot)$ the partial trace over all the systems except the $i$th subsystem. Notice that we are considering exact transformations for every $N$ and, as said before, we are not allowing fluctuations in the values of work[11]. Let us call $\mathcal{C}(\rho, N)$ the set of all the diagonal states which satisfy the partial trace condition of Eq. (8). We can now define the c-work of formation $\mathcal{W}_{form}(\rho, N)$ as the minimum work cost of this transformation over all the states in $\mathcal{C}(\rho, N)$:

$$\mathcal{W}_{form}(\rho, N) = \min_{\rho^{(N)} \in \mathcal{C}(\rho, N)} W_{form}(\rho^{(N)}). \qquad (9)$$

In what follows we will show that it is possible to find this minimum work cost and characterize a set of states that achieve this bound.

In order to carry out the minimization, first notice that the c-work of formation is always minimized by a state $\rho_{min}^{(N)}$ that is maximally mixed in each populated subspace of energy (see Supplementary Note 1 for details). Thus, one can reduce the set $\mathcal{C}(\rho, N)$, where the minimization is done, to a smaller subset of states. These states are such that $\lambda_{E,g} = p_E / g_N(E) \equiv \lambda_E$, where $p_E$ is the occupation of the subspace of energy $E \in \mathcal{E}_N$, $\mathcal{E}_N$ is the spectrum of the $N$-partite system and $g_N(\cdot)$ is the degeneracy. Therefore, each element of the subset is determined by just specifying the corresponding distribution $\lambda_E$. Notably, the minimization in Eq. (9) can be written as an optimization problem subject to linear constraints:

$$\min_{\{\lambda_E\}_{E \in \mathcal{E}_N}} k_B T \log\left[\max_E\{\lambda_E e^{\beta E} Z_S^N\}\right]$$
$$\text{s.t.} \sum_{E \in \mathcal{E}_N} g_{N-1}(E - E_d)\lambda_E = p_d \quad \forall d = 1, \dots, D \qquad (10)$$
$$\lambda_E \geq 0 \quad \forall E \in \mathcal{E}_N.$$

Moreover, this system of equations can easily be transformed into a linear optimization problem[42]. Both constraints define a bounded and non-empty convex set, and therefore there exists at least one optimal feasible solution. Since the optimization problem is linear there is an efficient algorithm, known as the simplex algorithm, that allows to solve the problem numerically. Furthermore, we will show how to fully characterize $\mathcal{W}_{form}(\rho, N)$ and the energy distribution $\lambda_E$ that solves the minimization problem for every local state $\rho$ and number of copies $N$.

For simplicity, we will first present our results for the particular case where each subsystem has dimension $D = 2$. However, our findings can be generalized to subsystems of arbitrary dimension, although the mathematical treatment is more involved. Without loss of generality we will consider $H_S = E_0|1\rangle\langle 1|$ as the Hamiltonian of each subsystem, and a general diagonal local state $\rho = (1-p)|0\rangle\langle 0| + p|1\rangle\langle 1|$. In this way, the local thermal

Gibbs state is defined as the state with $p = p_\beta$, where $p_\beta = (1 + e^{\beta E_0})^{-1}$, and partition function $Z_S = \text{tr}(e^{-\beta H_S})$. Our first result shows the analytical solution to the optimization problem of Eq. (10).

**Theorem 1.** Given an integer $N$ and a state $\rho$ which satisfies $[\rho, H_S] = 0$, there exists a subset of energies $\mathcal{E}_N^\rho \subseteq \mathcal{E}_N$, a constant $s \in (0, 1]$, and at most a single energy $\varepsilon \in \mathcal{E}_N$ such that the state $\rho_{\min}^{(N)}$ is defined by the distribution:

$$\lambda_E = \begin{cases} \frac{e^{-\beta E}}{\gamma} & \text{if } E \in \mathcal{E}_N^\rho \\ s\frac{e^{-\beta \varepsilon}}{\gamma} & \text{if } E = \varepsilon \\ 0 & \text{otherwise} \end{cases}, \quad (11)$$

with $\gamma$ a normalization constant. Furthermore, the work of formation and the extractable work of the optimal state $\rho_{\min}^{(N)}$ are given by

$$\mathcal{W}_{\text{form}}(\rho, N) = k_B T \log\left[\frac{Z_S^N}{\gamma}\right], \quad (12)$$

$$\mathcal{W}_{\text{ext}}(\rho, N) = k_B T \log\left[\frac{Z_S^N}{Z}\right], \quad (13)$$

respectively, where $Z$ is the partition function of a system in a thermal state at temperature $T$ with spectrum given by the set $\mathcal{E}_N^\rho \cup \{\varepsilon\}$, and $\gamma = Z - (1 - s)g_N(\varepsilon)e^{-\beta \varepsilon}$.

*Proof.* See Supplementary Note 2.

The structure of the optimal states is simple: except for the occupation of a single level with energy $\varepsilon$, $\rho_{\min}^{(N)}$ is a Gibbs thermal state over a reduced support of energies $\mathcal{E}_N^\rho$ which depends upon the local state $\rho$ and the number of copies $N$. The optimal states do have correlations that are the result of removing the population of some energy levels from the thermal state. Notice that a typical approximation that is usually done when one deals with large number of identical systems is similar to what is obtained in Eq. (11), i.e., discard tails of the energy distribution[4].

In this way, Eq. (12) gives the optimal work cost for creating a set of $N$ particles in a given local state. Example calculations of the c-work of formation per copy $\mathcal{W}_{\text{form}}/N$, the work of formation of a single copy $W_{\text{form}}$, and the amount of correlations in the optimal state for $N = 3$ are shown in Fig. 2a. The c-work of formation per copy lies below the work of formation of a single copy. The difference between these two curves is precisely the energy per copy that is saved in the process of formation due to the collective action. Figure 2b further stresses the difference between the c-work of formation per copy and the work of formation of a single copy. In fact, one can notice that there exist extreme cases, near the thermal state, where this ratio is minimal: $\mathcal{W}_{\text{form}}(\rho, N) = W_{\text{form}}(\rho)$. These states are such that the work of formation of a single copy is equal to the amount of work one should invest to obtain $N$ correlated copies, but on the other hand one cannot extract any deterministic work from them (for a more detailed explanation of these states, see Supplementary Note 4).

We have seen that the work of formation can be reduced if one acts collectively and creates correlations in the final state. This property appears when work is not allowed to fluctuate and thus the work of formation is greater than the free energy difference. However, not always the presence of correlations will help in the process. Notably, there is an upper bound on the amount of correlations that can be built up while reducing the work of formation:

$$\frac{1}{N}\mathcal{I}(\rho^{(N)}) \leq \beta \, \delta Q, \quad (14)$$

where $\delta Q \equiv W_{\text{form}}(\rho) - \Delta F(\rho)$ is called the dissipated work, the difference between the deterministic work of formation of a single copy with the free energy difference (see Supplementary Note 3 for a proof of the bound). Thus, correlations greater than $N\beta\delta Q$ are costly, since collective operations cannot outperform the single copy creation.

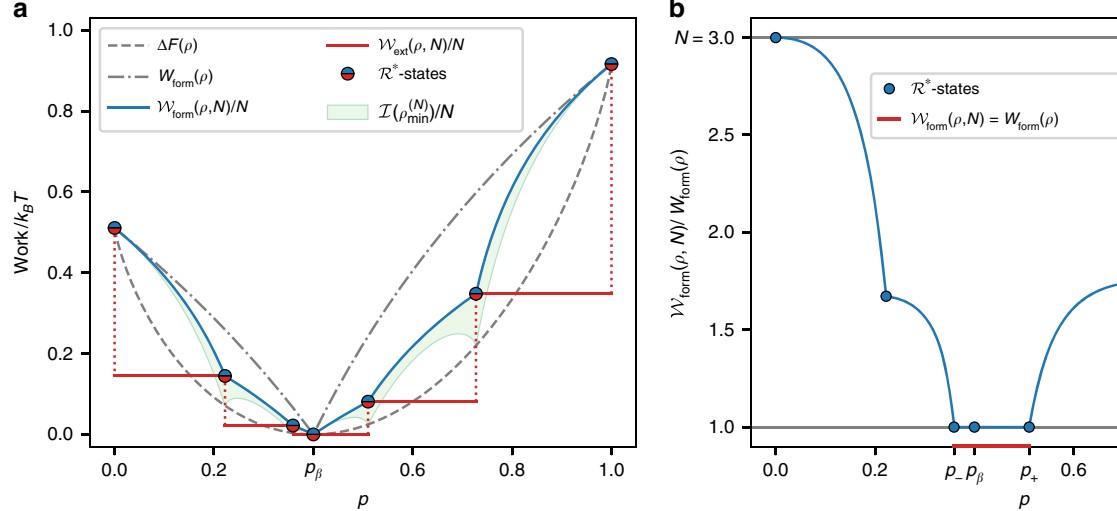

**Fig. 2** Minimum work of formation of three correlated copies for the different local qubit states. **a** Different measures of work as a function of the local qubit state, $\rho = (1 - p)|0\rangle\langle0| + p|1\rangle\langle1|$, parametrized by the excited state probability $p$. For one copy, the work of formation $W_{\text{form}}(\rho)$ (gray dash-dotted lines) is bigger than the standard free energy difference $\Delta F$ (gray dashed lines). On the other hand, the minimum work of formation of correlated copies (c-work of formation) per copy, $\mathcal{W}_{\text{form}}(\rho, N)/N$ (blue solid line), is smaller than or equal to $W_{\text{form}}(\rho)$ but still greater than $\Delta F(\rho)$. $\mathcal{W}_{\text{ext}}(\rho, N)/N$ (red solid line) is the extractable work per copy of the optimal correlated state $\rho_{\min}^{(N)}$. While in general all the correlated states are irreversible, $\mathcal{W}_{\text{ext}}(\rho, N) < \mathcal{W}_{\text{form}}(\rho, N)$, there are states (dots) whose $\rho_{\min}^{(N)}$ satisfies $\mathcal{W}_{\text{ext}}(\rho, N) = \mathcal{W}_{\text{form}}(\rho, N)$, this is the set of $\mathcal{R}^*$-states (reversible optimal states). The green region represents the total correlations per copy $\mathcal{I}(\rho_{\min}^{(N)})/N$ present in each multipartite state. **b** Ratio between the c-work of formation and the work of formation of a single copy. Creating the correlated state $\rho_{\min}^{(N)}$ costs less work than $N$ uncorrelated copies. There are some extreme cases (red) around the thermal state $p_\beta$ where the work of formation of a single copy equals the work of formation of $N$ correlated copies

**Reversibility.** A key result in the single-shot regime is the appearance of an intrinsic irreversibility: the extractable work is in general smaller than the work of formation. Thus, in general, one cannot extract the same amount of energy invested in the process of creation. However, it can be easily seen that there are families of states whose work of formation and extractable work coincide, and in this sense these states are reversible. Theorem 1 shows that, in fact, reversibility appears naturally in our framework. The states we define are such that in general the $c$-work of formation is greater than the extractable work, and the difference between these values is the irreversible work:

$$\mathcal{W}_{\mathrm{irr}}(\rho, N) = k_{\mathrm{B}} T \log[Z/\gamma]. \qquad (15)$$

For $s \approx 1$ the irreversible work is $\mathcal{W}_{\mathrm{irr}}(\rho, N) \approx (1-s) g_N(\varepsilon) e^{-\beta\varepsilon}/Z$. Remarkably, there is a subset of reduced states $\rho_k^{*(N)}$ for which their corresponding $\rho_{\mathrm{min}}^{(N)}$ is a thermal state over the reduced support $\mathcal{E}_N^\rho \cup \{\varepsilon\}$ (i.e., $s=1$). These states are such that their work of formation is equal to the extractable work, and in this sense they are strictly reversible, i.e., $\mathcal{W}_{\mathrm{irr}}(\rho_k^*, N) = 0$. On the other hand, the irreversibility increases as $s \to 0$.

We call the set of local states whose $\rho_{\mathrm{min}}^{(N)}$ are reversible $\mathcal{R}^*$-states. This set is composed by the states that match the break points in the curves of Fig. 2. In Supplementary Note 4 it is shown that there are $2N+1$ of such states $\rho_k^* = (1-p_k^*)|0\rangle\langle 0| + p_k^*|1\rangle\langle 1|$ with $k=1, 2, \ldots, 2N+1$ (see Fig. 2a). Furthermore, for these states the work (either of formation or extractable) can be expressed as

$$\mathcal{W}(\rho_k^*, N) = N\Delta F(\rho_k^*) + k_{\mathrm{B}} T \mathcal{I}_k, \qquad (16)$$

where $\mathcal{I}_k$ is the amount of total correlations present in the optimal state. Thus, work is the sum of two contributions: the classical value of work, given by the free-energy difference, plus the energy associated to the creation of correlations, but still $\mathcal{W}_{\mathrm{form}}(\rho_k^*, N) \leq N \, \mathcal{W}_{\mathrm{form}}(\rho_k^*)$. This shows that collective operations allow us to perform reversible transformations using deterministic work. Moreover, in the optimal process it is the energy of the correlations that fills the gap between the standard work of formation of independent copies and the $c$-work of formation (see Fig. 2a). Furthermore, as we will see below, these states have other interesting properties that will allow us to recover standard results from thermodynamics in the large $N$ limit.

**Finite-$N$ behavior and thermodynamic limit.** We have established that the $c$-work of formation represents the minimum amount of energy that is necessary to produce $N$ correlated copies of a state $\rho$ in a deterministic process. The natural question that follows is how these results behave as the number of copies increases. Figure 3a illustrates the behavior of the $c$-work of formation per copy for a few values of $N$. There, it is shown that the $c$-work of formation per copy approaches the free energy difference as $N$ increases. In addition, the set of $\mathcal{R}^*$-states increases linearly with $N$.

Further insight can be gained if one considers the density of reversible states. In Supplementary Note 4 we show that for any local state $\rho$ and $\varepsilon > 0$, there exists a number of copies $N = \mathcal{O}(1/\varepsilon)$ and an $\mathcal{R}^*$-state with density matrix $\rho^*(\varepsilon)$ that satisfies $\|\rho - \rho^*(\varepsilon)\| < \varepsilon$. This means that the set of $\mathcal{R}^*$-states is dense in the space of states with the local constraint (Eq. (8)). Moreover, since the irreversible work per copy goes to zero with $N$, for a large number of copies the process of formation is almost reversible. Thus, in the thermodynamic limit, we recover the standard results from thermodynamics:

**Theorem 2.** Let $\rho$ be any diagonal local state of a system S. Then

$$\frac{\mathcal{W}(\rho, N)}{N} \overset{N \to \infty}{\to} \Delta F(\rho) \qquad (17)$$

where $\mathcal{W}$ refers to either the $c$-work of formation or the extractable work of the optimal states, $\Delta F(\rho) \equiv F(\rho) - F(\tau_S)$ is the standard nonequilbrium free energy difference, and the rate of convergence is $\mathcal{O}(\log N/N)$.

*Proof.* See Supplementary Note 5.

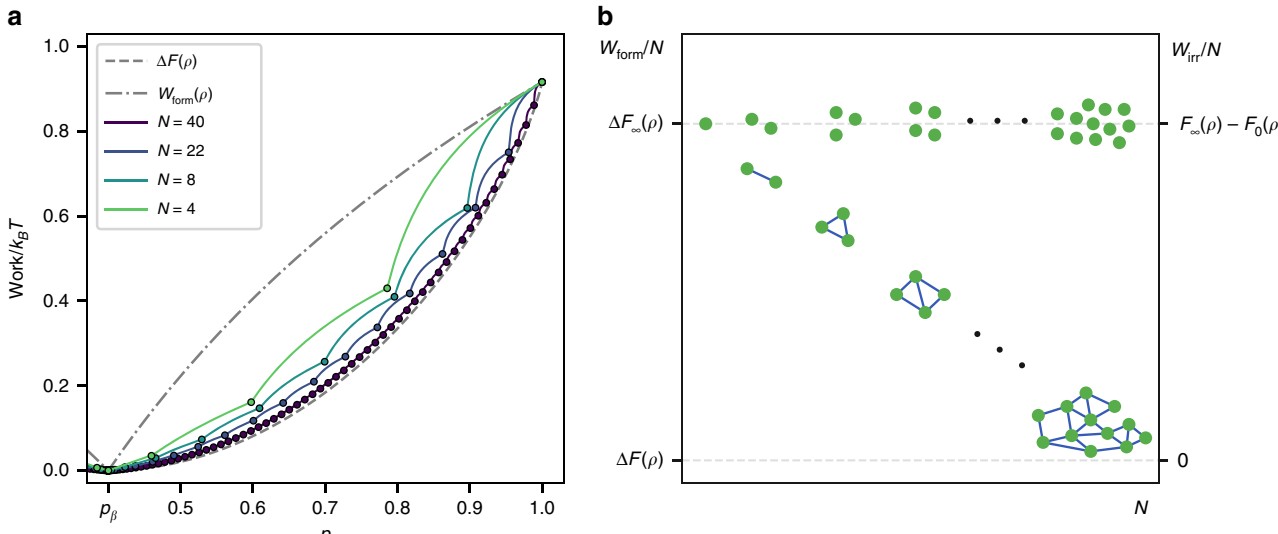

**Fig. 3** Asymptotic behavior of the minimum work of formation of correlated copies as the number of copies increases. **a** The minimum work of formation of correlated copies ($c$-work of formation) per copy $\mathcal{W}_{\mathrm{form}}/N$ is plotted in color for different values of $N$ (solid lines) as a function of the local reduced state, $\rho = (1-p)|0\rangle\langle 0| + p|1\rangle\langle 1|$, parametrized by the excited state probability $p$. As $N$ increases $\mathcal{W}_{\mathrm{form}}/N$ approaches the standard free energy difference $\Delta F$ (dashed gray line) for every state and the number of reversible states ($\mathcal{R}^*$-states) (represented by dots) increases linearly with $N$. In addition, for irreversible states the degree of irreversibility per copy, measured by the difference between the work of formation (solid colored line) and the extractable work (nearest point to the left), decreases with $N$. **b** Illustration of the results obtained for correlated copies. As $N$ increases the work of formation per copy approaches the free energy difference and furthermore the irreversible work per copy goes to zero; thus recovering reversibility

As it is illustrated in Fig. 3b by increasing the number of correlated copies we approach standard thermodynamics. Additionally, for large $N$ the total correlations in the optimal state are of order:

$$\mathcal{I}(\rho_{\min}^{(N)}) \sim \mathcal{O}(\log N), \qquad (18)$$

meaning that the amount of correlations per particle $\mathcal{I}(\rho_{\min}^{(N)})/N$ is negligible in the thermodynamic limit. These results establish that asymptotically the work per copy required to form $N$ correlated states is exactly what we expect when unbounded fluctuations are allowed, with an amount of correlations that increases sublinearly with the number of copies. Previous analysis of the thermodynamic limit[4,43], considered interconversion rates of product states in the limit of large number of particles using approximate transformations. Here, we consider locally exact transformations, and find the solution that ranges from small number of systems to arbitrary large ones. In this way, by obtaining the exact minimum work cost for every $N$ we could evaluate the deviation from standard results in every instance. More importantly, we have shown that in this approach the creation of correlations is the physical mechanism that leads to the emergence of the standard scenario.

**Generalizations**. These results were presented using the simplest example given by local systems of dimension $D = 2$. In fact, more complex systems can be considered by increasing $D$, and we can show that these ideas also hold for arbitrary local dimension $D$ (see Supplementary Note 6). The main difference with respect to the case $D = 2$ is that instead of having a single energy $\varepsilon$ (see Theorem 1), each optimal state $\rho_{\min}^{(N)}$ is obtained by considering a set of energies $\varepsilon_i$ and parameters $s_i \in (0,1]$, with $i = 1, ..., D-1$. The results concerning the thermodynamic limit have the same form.

Up to now, we have focused on the situation where all subsystems have the same Hamiltonian and same reduced state. A natural extension of our findings is to consider a non-symmetric case, where each subsystem is different. There, one can also show that correlations reduce the work of formation, and that the optimal state has a thermal-like distribution similar to the one in Theorem 1 (see Supplementary Note 7). Furthermore, by allowing correlations it is possible to recover standard results in the thermodynamic limit for a general configuration in our framework, that is for a set of different diagonal states with vector probabilities $p^{(i)}$ and Hamiltonians $H_i$ taken from an arbitrary distribution $\mathcal{D}$.

**Theorem 3**. Let $(p^{(1)}, E^{(1)}), (p^{(2)}, E^{(2)}), \dots, (p^{(N)}, E^{(N)}) \in \mathbb{R}_{\geq 0}^{2D}$ an i.i.d. sample with arbitrary distribution $\mathcal{D}$ and $\mathcal{W}_N$ the c-work of formation of a system with diagonal reduced states $\rho_i$ defined by the probability vector $p^{(i)}$ and Hamiltonian with energies $E^{(i)}$. Then,

$$\frac{\mathcal{W}_N}{N} \overset{N \to \infty}{\to} \langle \Delta F \rangle_{\mathcal{D}}, \qquad (19)$$

where the mean in $\Delta F$ is with respect to $\mathcal{D}$ and the convergence is almost surely.

*Proof*. See Supplementary Note 7.

In this case the c-work of formation $\mathcal{W}_N$ can be thought as a random variable since the state of the $N$ subsystems is chosen randomly following the distribution $\mathcal{D}$. For instance, if $\mathcal{D}$ has density $f$ taking values in $\Omega$, then

$$\langle \Delta F \rangle_{\mathcal{D}} = \int_{\Omega} \Delta F(p, E) f(p, E) dp \, dE. \qquad (20)$$

When dealing with copies the distribution is defined by $f(p, H) = \delta(p - \tilde{p})\delta(E - \tilde{E})$, where $\delta(\cdot)$ is the Dirac delta distribution. Thus,

our approach can be directly extended to more general settings including the asymmetric case.

## Discussion

We have presented a framework to study how the presence of correlations affects thermodynamic processes taking place in the single-shot regime. By first considering the formation of locally equivalent states, we show that the creation of correlated systems provides an advantage, since in this case the energetic cost of the process is lower than in the uncorrelated scenario. This is a feature that appears when fluctuations of work are constrained. Although we focused most of our analysis on the creation correlated of copies, we have shown that the same ideas could be extended to more general scenarios. Here, we have analyzed the creation of states that are diagonal in the energy eigenbasis. Creation of states with coherence in this scenario is strictly impossible, a source of coherences is required[4,15]. We think that these ideas could also be extended to this situation, as the amount of coherences can be reduced when one acts collectively. If independent copies require an amount of coherences of order $\mathcal{O}(N)$, a collective action reduces this requirement to $\mathcal{O}(\sqrt{N})$[4,15].

The description we provide is compatible with standard results of thermodynamics in the large $N$ limit. In fact, we have shown that this mechanism leads to the emergence of reversibility when the minimum work cost is considered. Unlike previous approaches, here we consider that the final state is correlated but all transformations and work extraction are exact and deterministic. Interestingly, we have also shown that an amount of correlations per copy that is vanishing small is sufficient to recover standard results in the large $N$ limit. Therefore, we can identify a physical mechanism, related with the creation of correlations, that allows to continuously approach standard results in the thermodynamic limit. Furthermore, we have also shown that classical results can also be recovered in the large $N$ limit with more general settings. We expect our work sheds light on the role of correlations in thermodynamic transformations of microscopic systems and its connection with the emergence of standard results.

## Data availability
The data that support the findings of this study are available from the corresponding authors upon request.

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

## Acknowledgements

The authors thank L. Masanes for useful comments, also P. Groisman and G. Acosta for discussions. F.C. and A.J.R. acknowledge support from CONICET, UBACyT, and ANPCyT.

## Author contributions

F.S., F.C., and A.J.R. contributed to all aspects of this work.
