## [Peer Review File · Nature Communications]

Reviewers' comments:

Reviewer #1 (Remarks to the Author):

The present manuscript studies whether and how correlations between identical copies of quantum states can be utilized as a thermodynamic resource. It is shown that the energetic cost of creating N copies of the same state is lower than N times the cost to create a single state, if one allows for correlations between the N states. It is then further shown that for small N the formation of correlations is thermodynamically irreversible, meaning that the energy required to create the correlations is larger than what can be extract from them. Finally, the authors find that in the limit of large N thermodynamic reversibility is recovered.

It is still a somewhat open question to what extend quantum resources can be employed to outperform statements and limitations of classical thermodynamics. In particular, quantum coherence and quantum correlations are promising candidates for practically relevant resources in quantum computation and in quantum heat engines. The present manuscript addresses this question with the quantum information theoretic framework for quantum thermodynamics. The analysis is mathematically sound and the narrative is clear. Unfortunately, the manuscript succumbs to the rather mathematical mind set of quantum information theory, which loses the practical and phenomenological spirit of thermodynamics. Thus, the present results seem to be mostly interesting for a specialized mathematical audience and not very illustrative for the broad audience of Nature Communications. In conclusion, I cannot recommend the manuscript to be accepted, at least not in its present form.

My recommendation is based on the following issues that the authors may find interesting to consider:

(i) Major issue: Like in many papers concerned with the quantum information theoretic approach to thermodynamics the physical motivation and applicability is somewhat questionable. It is certainly interesting to see that correlations can be a resource, but in what physical situation would that be relevant? When reading the manuscript I was trying to come up with an example. The best I could find would be a quantum computational setting. However, in this case one would neither want to assume that the environment is always thermal, nor would one would to assume that correlations are only built between the N copies and not between quantum states and environment, nor would the final quantum state always be diagonal in energy. Are there any other physical and experimental situations in which the new findings might become relevant?

(ii) Minor issues:

(a) On the first page, right column the authors talk about "creations in the final state". I am pretty sure that it should read "correlations".

(b) On the second page the authors outline the quantum information theoretic framework. As has been stated before, assuming that the environment is always in a Gibbs state is rather restrictive. For instance, such a setting negates the possibility of squeezed reservoirs or correlations between system and environment. Thus, the framework is not nearly as general as it is presented in the specialized literature.

(c) In a thermodynamic lingo there is only ONE second law of thermodynamics, which is comprised of several STATEMENTS of the second law for various situations. Unfortunately, Ref. [24] of the manuscript used a imprecise and obscure way of expressing their findings, by "naming" a set of inequalities "family of second laws". Quite regrettably Ref. [24]'s neologism is seeming to develop into a standing phrase in quantum information theory. I would highly encourage the authors to use a more precise language, which could avoid confusion and resentment from the statistical physics community.

(d) Without much motivation it is assumed that N copies of the same quantum state can be performed. For the less well-informed reader it might be worthwhile to comment on why the no-cloning theorem does not apply here.

Reviewer #2 (Remarks to the Author):

The authors of this manuscript work in the framework of resource theory applied to work extraction/work formation protocols under the framework known as thermal operation which applies energy preserving unitaries to a composite system with the aim of extracting or doing work. It is an approach which is favored by the quantum information cohort of the emerging field of thermodynamics of quantum systems.

The main result of the work is the realisation that the work cost of generating multiple copies of a state within the thermal operations framework is reduced by allowing correlations in the final state. This is a reasonably interesting result although I would have liked the authors to really discuss their notion of correlations more explicitly. I feel that the section after theorem 1 could be improved. The authors seem to think that it is obvious that these classes of states are correlated but perhaps this could be made more explicit by calculation of the total correlations versus work extraction in some numerical examples. I enjoyed the discussion of reversibility but I found the other theorem in the paper i.e. theorem 2 concerning the thermodynamic limit rather unsurprising. Perhaps I missed something but perhaps the authors could do more in saying why this is an interesting/new result?

Overall the paper is a nice contribution to the literature on resource theories of thermodynamics and should be interesting for the relatively small community who are studying these. Nevertheless it is certainly a topic which seems to be popular in Nature Communications.

The paper is well written I spotted some minor things which could improve the paper and the authors may want to consider - I list them in order in which I spotted them.

1. The authors might want to cite in the introduction a couple of the reviews in quantum thermodynamics e.g

<http://iopscience.iop.org/article/10.1088/1751-8113/49/14/143001/meta>

And

<https://www.tandfonline.com/doi/abs/10.1080/00107514.2016.1201896>

2. I also think it might be nice to think about a couple of recent papers showing how correlations play a non trivial role in work extraction outside of the resource theory paradigm

Eg

<https://journals.aps.org/prx/abstract/10.1103/PhysRevX.5.041011>

<https://www.nature.com/articles/s41534-017-0012-8>

<https://journals.aps.org/prl/abstract/10.1103/PhysRevLett.121.120602>

3. First page second column looks like you used the word "creations" probably mean "correlations"

4. General question - you are restricting to diagonal states - what happens if this is relaxed. I feel that this complication should be discussed.

5. Above equation 8 there is a typo "lineal" should be linear ?

6. I was a little confused about the notation in theorem 1 when you use "hat" for E and s - you might want to rethink that.

Reviewer #3 (Remarks to the Author):

Referee Report

The authors analyzed the following question: how much deterministic work one need to create n partite state out of Gibbs state, by thermal operations, provided that the reduced states of each single party is given by ρ . They consider quasi-classical case of state diagonal in an energy basis. They obtain several interesting results. First they realize that this is linear programming. Then they provide more or less explicit formula in the case of a qubit.

In general, they obtain that for each number of parties, there are some states, for which work of formation is equal to extractable work. The number of states grows with the number of parties.

In the limit of large number of copies, the work of formation approaches macroscopic value determined by free energy. In this way the authors have obtained free energy as a limit of microscopic quantities, without making smoothing (i.e. without allowing for some small error).

Perhaps the closest previous result is Ref [17], where the authors have shown, that if one can return catalyst in correlated state (while initially uncorrelated), then by allowing for small error, one can regain reversibility. Thus, correlations of the catalyst have been changed into work. In the present manuscript, there is no catalyst, and the change is exact. The work is taken from correlations.

They also show, that the amount of correlations (quantified by means of n -partite mutual information) increases only logarithmically with number of parties.

The claims of the papers are clearly novel and important. Full understanding deterministic work and the formalism of thermal operations is still a challenge, and the paper is an important contribution. The paper is clearly written, and appropriately discussed in the context of the previous literature.

I find the paper nice, elegant and for sure interesting for community of thermodynamics inspired by quantum information.

I am though not quite convinced, where such a paper should be published. Clearly, it deserves wide audience, but I am not sure whether it is enough important to publish it in Nat. Com. I leave this decision to the Editor. I am happy to answer some additional questions, if needed.

Here my specific comment:

- 1) In figures I would consider work perh $kT \ln 2$ rather just per kT . But this is up to authors.
- 2) After (3) in suppl mat: I would write "product" rather than "separable", because "separable" is a different meaning in entanglement theory.
- 3) I think in Eqs 6,7, 13 sum notation is incorrect. E.g. in Eq 6, let us consider the first term of the sum over i , i.e. let us take $i=1$. Then we have sum of over lambdas labeled by $d_1 d_2 \dots d_N$,

but only d_1 is summed, and it is not know, what value should take the other indices.

I believe there should be simply N sums, each over different d_i .

We would like to thank all the referees for the careful reading and useful remarks that certainly encouraged us to make an improved version with some new interesting results. In the revised version, we have made substantial changes in order to address all the concerns that have been risen. In this way, we have removed some technicalities that appeared in the main text, and dedicated more space to explain the assumptions which are used in our approach, as well as the physical consequences of our results. We are happy with the new version which, we think, has also become more accessible to a broad audience.

Before answering in detail to each point of the reports, we will first list the main changes we made, which are also highlighted in the manuscript text file.

1. We rewrote part of the introduction where we also include some new references.

2. In the subsection 'overview', now we start by explaining why it is relevant to study thermodynamics in the single-shot regime and its relation with the standard approach. We have also included a reference showing that the practical implementation of every thermal operation can be decomposed as a sequence of physically simple operations.

3. In the subsection "Work of formation of correlated copies" we include a new paragraph showing that the standard approach to thermodynamics, where arbitrary large fluctuations of work are allowed, tells us that the presence of correlations cannot be used to reduce the work cost of our task. This is done by explicitly writing the relation between work, total correlations and free energy.

4. After presenting the form of the optimal solution for the two dimensional case, we have added an in-depth discussion about the correlations present in the optimal states and how they can actually be used to decrease the work cost of the task we defined. Furthermore, we also derived a simple upper bound on the total amount of correlations that can be generated while reducing the work of formation. We have also added to the plot in Fig. 1 a) the amount of correlations of the optimal state. In the section subsection 'reversibility' we also include an equation with a simple explicit relation between the work of formation and the amount of correlations for reversible states.

5. In the subsection "Generalizations" we have added new results. First, we show the extension of Theorem 1 to the case of arbitrary different subsystems, i.e. having different Hamiltonians and local states. Second, we show a generalization of the results in the thermodynamic limit to sets of systems sampled from an arbitrary distribution. This is presented as Theorem 3.

6. We have rewritten part of the Discussion section and we have also included a paragraph about a possible generalization of our results to non block-diagonal states. In this case, even the definition of work of formation is an open problem, but from some particular examples we mention, we conjecture that our results could also be generalized to such a case.

7. In the Supplementary Material we include the proofs of our new results. Namely, a) how to find the bound on costly correlations, b) the generalization of our results to generic systems (not only copies), c) the generalization of the thermodynamic limit.

8. We have also changed some notation regarding: the total correlations, c-work of formation, and some definitions in theorem 1, as it was suggested by a Referee.

Reply to Reviewer #1:

We thank the Referee for their careful reading of the manuscript and useful remarks that have helped us make an improved version with some new results and more accessible to a broad audience. Below, we answer to the specific concerns/comments, and we hope to convince the Referee of the relevance of our results for small scale thermodynamics.

It is still a somewhat open question to what extent quantum resources can be employed to outperform statements and limitations of classical thermodynamics. In particular, quantum coherence and quantum correlations are promising candidates for practically relevant resources in quantum computation and in quantum heat engines. The present manuscript addresses this question with the quantum information theoretic framework for quantum thermodynamics. The analysis is mathematically sound and the narrative is clear. Unfortunately, the manuscript succumbs to the rather mathematical mind set of quantum information theory, which loses the practical and phenomenological spirit of thermodynamics.

We agree with the Referee that in the first version we had included some comments about technical aspects of our work that may have hidden the central point of our results. Therefore, we decided to rewrite some sections in order to address these concerns. In the new version we put more emphasis on the assumptions behind our approach, the connection with standard thermodynamics, and their physical consequences, trying to make it clearer for a broad audience. In our work we consider thermodynamic transformations in the single-shot regime where deterministic work is invested/extracted during the processes. This assumption imposes several restrictions in the allowed transformations. Understanding thermodynamics with limited fluctuations is especially important in order to address the behaviour of small-scale systems, as in this case fluctuations could be as large as the value of work. Moreover, it is of practical relevance for the description of small-scale heat engines. The aim of the paper is to study the role of correlations in this regime. Notably, within the standard framework, the creation of correlations is always costly. This is now explicitly stressed in the new version. Thus, as a consequence, one could imagine that correlations cannot be used to save energy in thermodynamic transformations. Here we show that this is not the case, in fact when fluctuations of work are constrained the presence of correlations can reduce the energetic cost of certain tasks. Furthermore, we showed that if one minimizes this work cost and takes the thermodynamic limit, it is possible to recover standard results. Hence, we could identify a physical mechanism that leads to the emergence of classical results. In the new version we show that, although correlations can be used to reduce the work cost of these tasks, creating an amount of correlations bigger than a given threshold is always costly. On the other hand, while the formalism we use comes from single-shot information theory, the implications of these results are rather general and the implementation of the transformations we considered can always be decomposed in simple physical operations as it is shown in Ref. [40].

My recommendation is based on the following issues that the authors may find interesting to consider:

(i) Major issue: Like in many papers concerned with the quantum information theoretic approach to thermodynamics the physical motivation and applicability is somewhat questionable. It is certainly interesting to see that correlations can be a resource, but in what physical situation would that be relevant? When reading the manuscript I was trying to come up with an example. The best I could find would be a quantum computational setting. However, in this case one would neither want to assume that the environment is always thermal, nor would one would to assume that correlations are

only built between the N copies and not between quantum states and environment, nor would the final quantum state always be diagonal in energy. Are there any other physical and experimental situations in which the new findings might become relevant?

In the paper we show that correlations allow to reduce the work of formation. We choose to talk about the generation of copies of a given state, though we now demonstrate that this feature is not limited to the creation of copies. Our first question was about finding the minimum work cost of creating a set of N particles with the same reduced state. This task may be relevant, for instance, when one tries to design thermodynamic cycles. In the introduction we tried to motivate our approach with a simple example, and we apologize if it was not too illustrative. However, one can notice that the creation of this type of multipartite states could also be associated to the starting point of a given quantum test, quantum communication protocol or quantum algorithm, where relevant quantities are evaluated by testing many single quantum systems prepared in the same state one at a time. For example, the preparation of a set of particles whose reduction is the identity state, could be done using different strategies that require different resources, one can construct a given cat-like multipartite state, a bunch of pairs of maximally entangled states, etc. Those preparations leads to different states which are locally equivalent. Of course, for practical applications the definition of reduced state would depend on how much information describes our system, or the type of task one wishes to do. For instance, if two particles are tested at the same time, correlations could affect the result of the experiment and thus the local state should be defined by specifying a product state of two particles. In the new version, we change the way in which we introduce the example to make it clearer. We think that from a practical point of view this is a relevant example. As we said, in the new version, we show that our results can be generalized to more general setting, not just the creation of copies, and of course one could also find a better application. However, as we underline before the aim of the paper is to show that the presence of correlations in this regime could allow us to perform tasks that differ from the standard approach. We believe, in this sense, that our results are of fundamental importance.

(ii) Minor issues:

(a) On the first page, right column the authors talk about "creations in the final state". I am pretty sure that it should read "correlations".

We have corrected this typo.

(b) On the second page the authors outline the quantum information theoretic framework. As has been stated before, assuming that the environment is always in a Gibbs state is rather restrictive. For instance, such a setting negates the possibility of squeezed reservoirs or correlations between system and environment. Thus, the framework is not nearly as general as it is presented in the specialized literature.

We are interested in studying transformations in contact with reservoirs at thermal equilibrium. In this context non-equilibrium reservoirs, such as squeezed reservoirs, are viewed as a useful resources since one can perform transformations that are not allowed with thermal equilibrium reservoirs. In fact, it can be shown that with correlated or squeezed reservoirs the efficiency of heat engines can surpass standard thermodynamic bounds (see for instance PRX 7, 031044 (2017)). Thus, transformations with squeezed thermal states could be considered by taking into account the energy invested in squeezing for instance. Regarding the second issue, correlations between system and environment are indeed allowed in this framework. The energy conserving interactions between system and reservoir can create correlations between them. Of course we are not considering initial correlated states between system and reservoir, since one wants to take into account every exchange of energy between

system and reservoir, but they are allowed to be created during the evolution. Thus, we can address thermodynamic transformations under similar conditions to standard thermodynamics but with constrained fluctuations of work.

(c) In a thermodynamic lingo there is only ONE second law of thermodynamics, which is comprised of several STATEMENTS of the second law for various situations. Unfortunately, Ref. [24] of the manuscript used a imprecise and obscure way of expressing their findings, by "naming" a set of inequalities "family of second laws". Quite regrettably Ref. [24]'s neologism is seeming to develop into a standing phrase in quantum information theory. I would highly encourage the authors to use a more precise language, which could avoid confusion and resentment from the statistical physics community.

We clarified this issue in the new version.

(d) Without much motivation it is assumed that N copies of the same quantum state can be performed. For the less well-informed reader it might be worthwhile to comment on why the no-cloning theorem does not apply here.

The no-cloning theorem states that one cannot design a physical device that makes copies of unknown states. Thus, it imposes a constraint in the allowed physical transformations, but does not prevent the generation of copies of previously defined states. Therefore, it does not apply in this approach.

We hope to have convinced the Referee of the importance of our work and finds the revised version suitable for publication.

Reply to Reviewer #2:

We thank the Referee for a careful reading, positive comments and helpful suggestions that stimulated us to find some new results. Below, we answer to each of the previous concerns.

The main result of the work is the realisation that the work cost of generating multiple copies of a state within the thermal operations framework is reduced by allowing correlations in the final state. This is a reasonably interesting result although I would have liked the authors to really discuss their notion of correlations more explicitly. I feel that the section after theorem 1 could be improved. The authors seem to think that it is obvious that these classes of states are correlated but perhaps this could be made more explicit by calculation of the total correlations versus work extraction in some numerical examples.

We thank the Referee for pointing this out, and we have taken into account this suggestion. In the new version we explain the notion of correlations, we also emphasize that in standard thermodynamics correlations are costly and our approach identifies a different feature that appears only in the single-shot regime. We actually rewrote the section after the theorem 1 where:

- a) we include the amount of correlations of the optimal states in Fig.1 a).
- b) we show that there is a simple bound on the amount of correlations that could be developed while reducing the work of formation.
- c) we include an explicit simple formula for the amount of correlations in the case of reversible states.

I enjoyed the discussion of reversibility but I found the other theorem in the paper i.e. theorem 2 concerning the thermodynamic limit rather unsurprising. Perhaps I missed something but perhaps the authors could do more in saying why this is an interesting/new result?

Theorem 2, and its generalization for arbitrary dimensional states, shows that the free energy is the energy per particle (in the thermodynamic limit) one should invest in the single-shot regime in order to create an ensemble of identical particles in the same reduced state. We demonstrate that this is achieved with an amount correlations that scales sublinearly with the number of systems, and in fact we show that such scaling is logarithmic. Additionally, it is shown that this transformation can be done reversibly. Thus, we recover classical results from thermodynamics within the same approach that rule small scale systems, provided a small amount of inner correlations are allowed.

Previous analyses of the thermodynamic limit in the resource theory were based for instance on the determination of the rate of interconversions, where formation (and distillation) of uncorrelated copies (product states) with approximate transformations were considered. These approximate transformations were studied from the beginning in the thermodynamic limit and mean that the final state is not the desired state but some (unknown) ε -close state. Furthermore, this approach also require the generation some additional exhaust state at the end of the protocol (see Ref[4]).

We understand that in light of the previous approaches and the analysis we showed, it would be expected that one can recover standard thermodynamic results in the large N limit. Nonetheless, we think that is not obvious that it is possible to obtain these results just by the addition of inner correlations. In fact, the approach and explanation we propose is fundamentally different from previous analyses, and it provides a new physical insight into a possible physical mechanism responsible of the emergence of standard thermodynamics from the single-shot scenario. We can

recall that if one starts by considering transformations in the single-shot regime, where the work of formation of every state is given by the Eq. 3, it is simple to see that if one considers the creation of uncorrelated copies a collective operation cannot lead to standard results in the thermodynamic limit. This is due to the fact the work of formation of uncorrelated copies is equal to the sum of work of formation single copies. Thus, an extra hypothesis must be included in the analysis. Theorem 2 states that it is indeed possible to recover standard results in the large N limit without invoking neither approximate energy conservation or transformations, nor the generation of exhaust states. Notably, one can arrive at this conclusion after performing the minimization of the work of formation over correlated states. On the other hand, notice that the local reductions of the optimal states we found are exact, and it is the collective action that leads to the creation of correlations that allows to reduce this work cost. Furthermore, we provide the exact analytical solution to the problem for every N, and we show that it converges to the conventional limit. This is something that was not present in previous approaches, where only convergence of an approximate state in the thermodynamic limit was studied even in the simplest cases. In summary, this section shows that a possible explanation to the appearance of standard results in the thermodynamic limit is through the presence of correlations between subsystems, that also become vanishing small as one increases the number of subsystems. As a byproduct of the generalization of our findings to arbitrary systems we could also extend the results in thermodynamic limit for sets of systems sampled from an arbitrary distribution, this is now included in the new version.

The paper is well written I spotted some minor things which could improve the paper and the authors may want to consider - i list them in order in which i spotted them.

1. The authors might want to cite in the introduction a couple of the reviews in quantum thermodynamics e.g

<http://iopscience.iop.org/article/10.1088/1751-8113/49/14/143001/meta>
<https://www.tandfonline.com/doi/abs/10.1080/00107514.2016.1201896>

2. I also think it might be nice to think about a couple of recent papers showing how correlations play a non trivial role in work extraction outside of the resource theory paradigm

Eg

<https://journals.aps.org/prx/abstract/10.1103/PhysRevX.5.041011>

<https://www.nature.com/articles/s41534-017-0012-8>

<https://journals.aps.org/prl/abstract/10.1103/PhysRevLett.121.120602>

We thank the Referee for pointing this out, we have included all these references in the new version.

3. First page second column looks like you used the word “creations” probably mean “correlations”

5. Above equation 8 there is a typo “lineal” should be linear ?

We have corrected these typos.

4. General question - you are restricting to diagonal states - what happens if this is relaxed. I feel that this complication should be discussed.

In this work we consider only block-diagonal states, strictly speaking the creation of non-diagonal states is not possible using thermal operations, because given that they commute with the Hamiltonian of the system they cannot create coherences between energy levels. One needs to add a coherence source in order to achieve this transformation, and fully characterizing the cost of formation

for non diagonal states remains an open question. However, we think that one can still obtain an advantage by adding correlations, since it has been shown that the amount of coherence can be reduced if one acts collectively. We now include a comment on this issue in the discussions section.

6. I was a little confused about the notation in theorem 1 when you use “hat” for E and s - you might want to rethink that.

We thank again to the Referee to pointing this out, we have changed notation in the new version.

Reply to Reviewer # 3:

We thank the Referee for the careful reading and very positive comments. We have taken into account their main concern and we have rewritten several parts of the manuscript. In the revised version, we now stress that the presence of correlations affects small-scale thermodynamics in a way that is different from what is expected with a standard analysis and is only present when the fluctuations of work are constraint, we have extended some of the results to more general situations, and we also emphasize the fundamental importance of our work in understanding the emergence of classical thermodynamics from the small-scale regime. Below we answer to the specific remarks.

1) In figures I would consider work perh kT In 2 rather just per kT . But this is up to authors.

We have considered the suggestion. If the Referee agrees, we prefer to keep kT since now we also include the amount of correlations in the same plot.

2) After (3) in suppl mat: I would write “product” rather than “separable”, because “separable” is a different meaning in entanglement theory.

Now we use “product” instead of “separable”.

3) I think in Eqs 6,7, 13 sum notation is incorrect. E.g. in Eq 6, let us consider the first term of the sum over i , i.e. let us take $i=1$. Then we have sum of over λ s labeled by $d_1 d_2 \dots d_N$, but only d_1 is summed, and it is not know, what value should take the other indices. I believe there should be simply N sums, each over different d_i .

We thank the Referee for pointing this out, in fact the notation we used was a bit misleading, thus we have changed it accordingly.

REVIEWERS' COMMENTS:

Reviewer #1 (Remarks to the Author):

In my previous report on the manuscript I already mentioned that the analysis is interesting with the mathematical framework of quantum information theory. In their revisions, the authors have tried to address all issues raised by the referees. Unfortunately, I am still not convinced that the analysis is more than a mathematical framework with limited experimental, i.e., practical consequences.

However, as has been pointed out by another referee, Nature Communications has published articles of similar quality before. Although I still think that the manuscript would be better suited for a more specialized journal, Nature Communication might actually be exactly this specialized journal for quantum information theoretic approaches to mathematical questions (motivated by thermodynamics).

Reviewer #2 (Remarks to the Author):

I felt that the authors of the paper have made a major effort to improve the paper and it is now ready for publication.

Reply To Referees

We again thank all the referees for their helpful comments and provide responses to each one of their new comments below.

Regards,
The Authors.

Reply to Reviewer #1:

In my previous report on the manuscript I already mentioned that the analysis is interesting with the mathematical framework of quantum information theory. In their revisions, the authors have tried to address all issues raised by the referees. Unfortunately, I am still not convinced that the analysis is more than a mathematical framework with limited experimental, i.e., practical consequences.

However, as has been pointed out by another referee, Nature Communications has published articles of similar quality before. Although I still think that the manuscript would be better suited for a more specialized journal, Nature Communication might actually be exactly this specialized journal for quantum information theoretic approaches to mathematical questions (motivated by thermodynamics).

We thank referee for their comments throughout the review process which have helped produce a better and more accessible article to a broader audience.

Reply to Reviewer #2:

I fell that the authors of the paper have made a major effort to improve the paper and it is now ready for publication.

No further issues were raised and we again thank the referee.